# Determination of the Most Suitable Cut-Off Point of the Cervical Foraminal Cross-Sectional Area at the C5/6 Level to Predict Cervical Foraminal Bony Stenosis

**DOI:** 10.3390/tomography11060067

**Published:** 2025-06-10

**Authors:** Joohyun Lee, Jee Young Lee, Keum Nae Kang, Jae Ni Jang, Sukhee Park, Young Uk Kim

**Affiliations:** 1Department of Anesthesia and Pain Medicine, CHA Ilsan Medical Center, CHA University, Ilsan 10414, Republic of Korea; hightothesky1004@gmail.com; 2Department of Korean Medicine, Integrative Cancer Center, CHA Ilsan Medical Center, CHA University, Ilsan 10414, Republic of Korea; happiade@chamc.co.kr; 3Department of Anesthesiology and Pain Medicine, National Police Hospital, Seoul 05715, Republic of Korea; cleanbinu@gmail.com; 4Department of Anesthesiology and Pain Medicine, International St. Mary’s Hospital, Catholic Kwandong University, Incheon 22711, Republic of Korea; sayhotcom@naver.com (J.N.J.); appealex@gmail.com (S.P.)

**Keywords:** cervical foraminal bony stenosis, cervical spine MRI, cross-sectional area, diagnostic cut-off, C5/6 level, radiculopathy

## Abstract

**Background**: Cervical foraminal bony stenosis (CFBS) is a common degenerative spinal condition that causes radicular pain and functional impairment in the upper extremities. Accurate and objective diagnosis of CFBS remains challenging due to the absence of standardized morphometric criteria. This study aimed to determine an optimal cut-off value for the cervical foraminal cross-sectional area (CFCSA) at the C5/6 level as a diagnostic indicator of CFBS. **Methods**: We conducted a retrospective case-control study including 154 patients aged 50 years or older with clinically and radiologically confirmed CFBS and 150 age-matched asymptomatic controls. Cervical spine magnetic resonance imaging (MRI) was performed in all subjects and CFCSA measurements were obtained from sagittal T2-weighted images using a standardized protocol. Group differences were analyzed using *t*-tests and diagnostic performance was assessed using receiver operating characteristic (ROC) curve analysis. **Results**: The mean CFCSA was significantly lower in the CFBS group (25.65 ± 7.19 mm^2^) compared to the control group (43.00 ± 8.38 mm^2^; *p* < 0.001). ROC analysis identified a CFCSA threshold of 33.02 mm^2^ as the optimal cut-off point for predicting CFBS, yielding a sensitivity of 86.4%, a specificity of 86.7%, and an area under the curve (AUC) of 0.94 (95% CI: 0.91–0.96). **Conclusions**: These findings suggest that CFCSA is a robust and reproducible morphological parameter for evaluating foraminal stenosis. The proposed cut-off may enhance diagnostic accuracy and aid in clinical decision-making for patients presenting with C6 radiculopathy. However, given this study’s retrospective, single-center design, further validation through multicenter, prospective studies across multiple cervical levels is warranted.

## 1. Introduction

Cervical foraminal bony stenosis (CFBS) is a frequently encountered condition in clinical practice and represents a significant source of pain, discomfort, and functional impairment involving the neck, shoulders, and upper extremities [1,2,3]. This clinical entity arises from structural narrowing of the intervertebral foramina, which serve as critical passageways for cervical nerve roots exiting the spinal column [4]. As these foramina become constricted, patients may present with a spectrum of neurologic symptoms, ranging from localized neck pain to radiating arm pain, numbness, paresthesia, and tingling sensations that often extend distally along the upper limbs [1,4,5]. In more advanced or severe cases, this nerve root compromise can progress from mere sensory disturbances to full-fledged cervical radiculopathy, where irritation or inflammation of the nerve roots within the narrowed foramina not only heightens pain and sensory deficits but may also manifest as motor weakness [1,2,5]. Such motor involvement can substantially affect a patient’s daily activities, quality of life, and capacity to engage in routine tasks [2,6].

In terms of imaging evaluation, magnetic resonance imaging of the cervical spine (CS-MR) has long been recognized as the modality of choice for early detection and detailed assessment of the morphological changes associated with CFBS. Its superior soft-tissue contrast and multiplanar capabilities enable clinicians to visualize not only the bony and cartilaginous structures but also the spinal cord, intervertebral discs, nerve roots, and surrounding soft tissues with exceptional clarity [3,4,5,6,7,8]. By delineating these anatomical relationships, CS-MR greatly assists in guiding the selection of appropriate therapeutic strategies. In clinical practice, differentiation of CFBS from other peripheral neuropathies or musculoskeletal conditions that present with similar pain and paresthesia is essential to ensuring accurate diagnoses and tailored interventions. Such differentiation is often achieved through CS-MR findings correlated with clinical presentation [9]. Indeed, treatment decisions—ranging from conservative management with physical therapy and medications to interventional procedures or surgical decompression—are frequently informed by the presence, severity, and distribution of neurological deficits observed on imaging [10]. Thus, the importance of precise and reliable radiological evaluation cannot be overstated.

CFBS is anatomically and pathophysiologically linked to various degenerative or structural changes that compromise the patency of the neural foramina. Factors such as loss of intervertebral disc height, herniated cervical discs, and the formation of osteophytes or bony spurs can collectively or individually contribute to foraminal encroachment and reduced space for the cervical nerve roots [11,12]. Accumulating evidence suggests that the degree of cervical foraminal narrowing is directly related to the severity of clinical symptoms and the progression of CFBS [1,3,7]. Given the complexity of cervical anatomy and the multifactorial nature of foraminal stenosis, a comprehensive diagnostic approach often incorporates multiple imaging modalities. While CS-MR provides excellent soft-tissue contrast, complementary imaging such as computed tomography (CT) and plain radiography (X-rays) can offer additional insights, particularly regarding the osseous aspects of cervical spine pathology [11,13]. By synthesizing these different imaging perspectives, clinicians can achieve a more holistic understanding of the structural changes at play.

Among the degenerative alterations contributing to CFBS, facet joint hypertrophy stands out as a particularly relevant mechanism [4,5,7]. Over time, degenerative changes and increased biomechanical stress on the facet joints can lead to their enlargement, further decreasing the available space within the intervertebral foramina. According to An et al., this degenerative hypertrophy often results in a reduction of foraminal dimensions and mechanical compression of the nerve roots, intensifying pain and functional limitations [14]. However, despite the recognition of various contributory factors, determining the exact structural cause of nerve root compression at frequently affected levels—such as C5/6—remains a challenging task. This difficulty arises from the lack of a precise, quantifiable parameter that can directly correlate morphological changes in the cervical foramina with clinical manifestations of nerve root compression. Despite multiple grading systems, the absence of a validated, quantitative imaging criterion continues to hinder consistent diagnosis. Defining such a parameter could reduce diagnostic subjectivity and facilitate more standardized clinical assessment—particularly through morphometric indicators such as cross-sectional foraminal area (CFCSA).

In recent years, growing interest has centered on identifying morphological parameters that accurately reflect the spatial dimensions of the intervertebral foramina. Among these potential indicators, the cervical foraminal cross-sectional area has emerged as a promising candidate for quantifying the degree of foraminal narrowing and its clinical relevance [14,15]. Since CFBS primarily results from degenerative alterations that reduce foraminal dimensions, understanding the role of CFCSA is logically of paramount importance. Yet, few investigations have directly examined the relationship between reduced CFCSA and the risk or severity of CFBS. Notably, Knapik et al. reported that patients with CFBS typically exhibit smaller intervertebral foraminal cross-sectional areas compared to asymptomatic controls, suggesting a strong link between diminished CFCSA and a heightened probability of developing CFBS [16]. However, while their findings underscore the importance of foraminal dimensions, no specific threshold value for CFCSA was established to serve as a robust diagnostic criterion, leaving clinicians without a clear quantitative benchmark. Grading systems, such as those proposed by Kim et al. and Park et al., primarily assess relative nerve root width or the extent of perineural fat obliteration, without providing a direct quantification of foraminal space. Furthermore, these methods are often subject to considerable interobserver variability, which may limit their broader clinical applicability [2,6]. This highlights the need to establish a quantitative threshold for CFCSA that can serve as a standardized diagnostic parameter in clinical practice.

This study aimed to assess the diagnostic utility of CFCSA in identifying CFBS and to establish a clinically applicable cut-off value. We hypothesized that CFCSA would serve as a reliable morphological marker to differentiate CFBS from normal anatomy. To the best of our knowledge, this is the first investigation designed to quantify the relationship between CFCSA and CFBS using well-defined imaging criteria and a controlled comparative analysis. By establishing a reliable CFCSA threshold, we seek to enhance the diagnostic accuracy for CFBS and support the development of more effective, individualized treatment strategies. In doing so, we hope to advance the current understanding of CFBS and provide clinicians with a valuable diagnostic tool to better predict and manage this debilitating condition.

## 2. Materials and Methods

### 2.1. Patient Selection

This study obtained formal approval from the Institutional Review Board (IRB) of the University of Catholic Kwandong, College of Medicine, located in South Korea, ensuring that all research activities adhered to established ethical standards. The IRB assigned the protocol number IS18RISI0047 to this investigation, thereby documenting its compliance with institutional, national, and international guidelines for the responsible conduct of research.

To explore the relationship between cervical foraminal narrowing and CFBS, we conducted a retrospective review of medical records and imaging studies for patients who underwent CS-MR imaging between June 2017 and January 2019 (Figure 1). These patients had all been diagnosed with CFBS, a pathological narrowing of the cervical neural foramen that can lead to radicular symptoms.

In order to focus our analysis on a well-defined and clinically relevant subgroup, we applied the following inclusion criteria:(1)Age ≥ 50 years(2)Symptomatic C6 radiculopathy based on clinical evaluation(3)Most severe foraminal narrowing observed at the C5/6 level on MRI(4)Underwent cervical spine MRI (CS-MR) within 6 months of initial CFBS diagnosis

To ensure the reliability and clinical relevance of our findings, we also applied the following exclusion criteria, aiming to eliminate confounding conditions or anatomical variations that could impact imaging interpretation:(1)Presence of cervical infections, trauma, or tumors(2)Moderate to severe central canal stenosis at cervical levels other than C5/6(3)History of cervical spine surgery(4)Evidence of cervical myelopathy(5)Presence of cervical cysts or syringomyelia

After applying these criteria, a total of 154 patients met all inclusion parameters and were thus categorized into the CFBS group for further analysis. To ensure diagnostic consistency and accuracy, all CFBS cases were independently verified by two experienced, board-certified neuroradiologists who reviewed the imaging studies and clinical records. Their consensus confirmation helped maintain a high standard of diagnostic rigor. Two neuroradiologists (each with over 10 years of experience) independently measured CFCSA. Inter- and intra-observer reliability were assessed using intraclass correlation coefficients, yielding values of 0.92 and 0.91, respectively.

As a reference standard for comparison, we also included 150 asymptomatic individuals who underwent CS-MR imaging as part of routine medical check-ups during the same study period. These healthy controls had no clinical or radiological evidence of cervical foraminal narrowing. For both CFBS patients and control subjects, the CFCSA was measured at the C5/6 foramen. By focusing on a single anatomical level and comparing measurements between symptomatic patients and controls, we aimed to objectively assess whether reduced CFCSA correlates with the presence of CFBS.

### 2.2. Imaging Protocols

We employed sagittal T2-weighted CS-MR imaging to measure CFCSA in both patients with clinically and radiologically confirmed CFBS and in healthy controls without symptoms or imaging evidence of foraminal stenosis. All CS-MR imaging was performed using MRI scanners, including 3.0-T systems (Philips Achieva, Philips Healthcare, Best, The Netherlands) and a 1.5-T system (Siemens Avanto, Siemens Healthcare, Erlangen, Germany). Only data obtained from 3.0-T MRI scanners were included in this study. Sagittal T2-weighted images were obtained with the following parameters: slice thickness = 2.0 mm, interslice gap = 0.4 mm, repetition time (TR) = 3740 ms, echo time (TE) = 87 ms, field of view (FOV) = 240 × 240 mm, matrix size = 448 × 314, and echo train length (ETL) = 15. Axial T2-weighted images were acquired with the following parameters: slice thickness = 3.0 mm, interslice gap = 0.3 mm, TR = 3000–4000 ms, TE = 90–120 ms, FOV = 180 × 180 mm, and matrix size = 320 × 256.

### 2.3. Image Measurements

To quantify the degree of foraminal narrowing and assess its relationship to CFBS, we reviewed sagittal T2-weighted CS-MR images, focusing on the most stenotic foraminal lesions at the C6 level. Using a picture archiving and communication system (INFINITT PACS, Incheon, Korea), the two neuroradiologists were blinded to the clinical information and group allocation (CFBS vs. control) and the images were reviewed in a randomized order to minimize interpretation bias. The measurement process involved carefully tracing the contours of the cervical neural foramen on the sagittal T2-weighted images (Figure 2). All sagittal CFCSA measurements were cross-referenced with axial T2-weighted images to ensure accurate localization of the foraminal level. Only high-quality images without significant artifacts were included. By capturing the precise dimensions of the foraminal region, these CFCSA measurements provided a quantitative parameter that could be directly compared between the CFBS patient group and the asymptomatic control group. Ultimately, our goal was to identify a CFCSA threshold that might serve as a diagnostic benchmark, thereby refining the evaluation and management of patients presenting with CFBS. Specifically, the CFCSA was delineated on sagittal T2-weighted images by manually tracing the bony margins of the neural foramen at the C5/6 level, using the following anatomical landmarks:Superior boundary: Inferior margin of the C5 pedicleInferior boundary: Superior margin of the C6 pedicleAnterior boundary: Posterior vertebral body and uncovertebral jointPosterior boundary: Facet joint complex and lamina

To enhance accuracy and ensure appropriate level localization, all sagittal measurements were cross-referenced with axial T2-weighted images. Only slices in which the full vertical extent of the foramen was clearly visible without motion artifacts were used for analysis. Discrepancies exceeding 5% in CFCSA measurements were resolved by joint review and consensus between the two neuroradiologists.

### 2.4. Statistical Analysis

To compare CFCSA values between the CFBS group and the asymptomatic control group, an unpaired *t*-test was performed. The threshold for statistical significance was set at *p* < 0.05. To assess age-related differences in CFCSA within the overall study population, subjects were stratified into three age groups: 50–59, 60–69, and ≥70 years. One-way analysis of variance (ANOVA) was used to compare CFCSA measurements across these age groups. Normality of distribution was assessed using the Shapiro–Wilk test prior to conducting parametric analyses. Furthermore, the diagnostic performance of CFCSA measurements was thoroughly evaluated using receiver operating characteristic (ROC) curve analysis. This included the determination of key diagnostic metrics such as cut-off values, sensitivity, specificity, and the area under the curve (AUC), which were reported along with their corresponding 95% confidence intervals (CIs). These metrics provided a comprehensive understanding of the ability of CFCSA values to distinguish between diagnostic categories. All results were expressed as mean ± standard deviation (SD). Statistical analyses were performed using IBM SPSS Statistics for Windows, Version 22.0 (IBM Corp., Armonk, NY, USA).

## 3. Results

All continuous variables satisfied the assumption of normality. The baseline characteristics of the control and CFBS groups are presented in Table 1. There were no statistically significant differences in demographic characteristics between the groups. The mean CFCSA was significantly smaller in the CFBS group (25.65 ± 7.19 mm^2^) compared to the control group (43.00 ± 8.38 mm^2^, *p* < 0.001).

When stratified by age, the mean CFCSA in the normal group was 42.71 ± 8.42 mm^2^ for individuals aged 50–59 years, 44.41 ± 7.78 mm^2^ for those aged 60–69 years, and 38.67 ± 10.96 mm^2^ in the 70–79 age group (Table 2).

There was no statistically significant association between age and CFCSA in the normal group (F = 1.44, df = 2, *p* = 0.240). In the CFBS group, the mean CFCSA values were 24.86 ± 7.10 mm^2^ for the 50–59 age group, 26.59 ± 7.51 mm^2^ for those aged 60–69 years, and 28.75 ± 6.32 mm^2^ for individuals aged 70–81 years (Table 2). Similarly, no significant age-related changes in CFCSA were found within the CFBS group (F = 2.464, df = 2, *p* = 0.89).

To evaluate the diagnostic utility of CFCSA for predicting CFBS, receiver operating characteristic (ROC) curve analysis was conducted. The optimal cut-off value for CFCSA was determined to be 33.02 mm^2^, which demonstrated a sensitivity of 86.4%, a specificity of 86.7%, and an area under the curve (AUC) of 0.94 (standard error = 0.0128; 95% CI, 0.91–0.96) (Figure 3).

## 4. Discussion

CFBS, which emerges from the progressive narrowing of the nerve root pathways passing through the cervical vertebral foramina, is a frequently encountered clinical condition known to cause considerable discomfort and functional impairment. As the foramina become constricted, patients may present with a diverse array of symptoms, ranging from sharp, radiating pain in the neck and arm to sensory disturbances such as numbness, tingling sensations, or paresthesia spreading into the upper extremities. In more advanced cases, as the narrowing intensifies, motor involvement may become evident, leading to weakness in the affected limb and further diminishing the patient’s quality of life [15,17,18]. Given the complexity of cervical spine pathologies, a precise clinical diagnosis of CFBS relies heavily on a combination of neurological evaluations and imaging studies. Within this diagnostic framework, CS-MR stands out as a key modality that provides detailed visualization of both soft tissue and bony structures, thereby enabling physicians to distinguish CFBS from other potential causes of peripheral neuropathy and to develop tailored treatment strategies [19,20].

A critical aspect of accurately diagnosing and managing CFBS involves recognizing the structural changes that underlie foraminal narrowing. Such narrowing is often attributed to degenerative phenomena, including the formation of osteophytes and the occurrence of lateral disc herniation—both of which can intrude upon the foraminal canal and impinge on the exiting nerve roots [1]. In an effort to standardize the evaluation of CFBS severity, various grading systems have been introduced in the literature. For example, Kim et al. proposed a classification scheme based on axial T2-weighted CS-MR images, categorizing stenosis into three distinct grades according to the foramen width and the ratio of the narrowest width to the extraforaminal nerve root [19]. Park et al. offered an alternative approach, employing oblique cervical images to assess the degree of perineural fat obliteration and further refine the characterization of foraminal stenosis [8,20]. Despite these efforts to objectify CFBS assessment, discrepancies between different grading systems frequently arise in clinical practice, calling their reliability into question and underscoring the need for a more universally applicable and quantifiable parameter.

We hypothesized that previous grading methods may have overlooked a crucial morphological factor: the direct measurement of the cervical foraminal cross-sectional area (CFCSA). From a biomechanical and degenerative perspective, repetitive cervical vertebral movements, coupled with age-related changes, can lead to hypertrophic alterations in the foramina [15]. Osteophyte formation, lateral disc protrusions, and facet joint overgrowth collectively contribute to a reduction in CFCSA, effectively encroaching upon the space through which nerve roots travel [21]. Among these degenerative changes, stenosis at the C5/6 level is particularly common, as highlighted by Ko et al., who reported that this specific site accounts for approximately 24.66% of CFBS cases [11]. Despite growing recognition of these underlying mechanisms, no standardized, widely accepted method currently exists to differentiate among the various potential causes of C6 nerve root damage originating from medial stenotic lesions at the C5/6 level.

To address this diagnostic gap, our study sought to determine the clinical utility and diagnostic reliability of CFCSA as a morphological parameter for identifying CFBS. By systematically analyzing CS-MR images, we compared intervertebral neural foramina measurements between patients suffering from CFBS and asymptomatic control subjects. The working premise was that degenerative stenotic alterations of the foramina drive the development of CFBS and that directly quantifying the CFCSA would allow for a more objective and reproducible measure of stenosis severity.

The primary morphological parameter in our study, the cervical foraminal cross-sectional area (CFCSA), demonstrated a statistically significant difference between groups, with a proposed cut-off value of 33.02 mm^2^. This parameter showed strong diagnostic performance in identifying CFBS, with a sensitivity of 86.4%, specificity of 86.7%, and an AUC of 0.94 (95% CI: 0.91–0.96). In comparison, An et al. proposed a cut-off value of 113.14 mm^2^ for the cervical facet joint area (CFJA) to predict foraminal stenosis in patients aged 50 and older at the C5/6 level, reporting a sensitivity of 70.6%, specificity of 68.6%, and an AUC of 0.72 (95% CI: 0.66–0.77) [15]. While both studies adopted a cross-sectional area-based diagnostic approach using ROC curve analysis, their measurement focused on facet hypertrophy rather than direct evaluation of foraminal narrowing. Moreover, the diagnostic performance of CFCSA in our study was notably superior, reinforcing its potential utility as a more precise and reproducible indicator of CFBS. Notably, we focused our analysis on individuals aged 50 years and older. This decision was guided by evidence suggesting that osteoarthritic changes in the cervical spine are minimal before the age of 50 and tend to advance with increasing age [22]. We interpret these results as evidence that repetitive mechanical stress on the cervical spine—stemming from routine activities involving extension, flexion, and rotation—facilitates the morphological changes responsible for foraminal narrowing. In response to instability and ongoing abrasion, osteophytes may form as a compensatory mechanism, ultimately reducing the CFCSA and exacerbating nerve root compression [15,21]. Consistent with this view, previous studies have reported that repetitive mechanical loading can augment osteophyte volume, leading to a further decrease in CFCSA [22,23,24,25]. Additionally, degenerative disc disease likely intensifies the biomechanical stresses on the foramina, placing osteophyte formation and facet joint hypertrophy at the forefront of structural factors contributing to CFBS [15,21].

Despite the promising implications of our study, several limitations must be acknowledged. First, the relatively small sample size necessitates further research involving larger cohorts to validate our findings and ensure their generalizability. In particular, the small number of participants in the older age subgroups (e.g., 70–79 years) limits the statistical power of age-stratified analyses and may have contributed to the lack of significant findings in age-related comparisons. Second, technical factors such as slice orientation and the 2.0-mm thickness of sagittal T2-weighted CS-MR images may have introduced variability in CFCSA measurements. Additionally, although CFBS can occur at multiple cervical levels, our investigation focused exclusively on the C5/6 foramen due to the high incidence of nerve root damage at this site. Third, these findings are limited to the C5/6 level and cannot be generalized to other cervical levels without further investigation. Additionally, the inclusion of only symptomatic C6 radiculopathy patients with C5/6 foraminal narrowing limits the generalizability of these findings to other cervical levels or other causes of CFCSA reduction. Fourth, CT imaging was not employed to further differentiate the etiologies of foraminal narrowing, such as facet joint hypertrophy or disc herniation, which may impact the interpretation of the underlying cause of CFCSA reduction. Fifth, other demographic factors, such as gender, body size, or body mass index, were not analyzed in this study. These factors may potentially influence CFCSA and warrant further investigation. Sixth, while CFCSA served as our primary morphological parameter, other structural indicators—such as lateral mass hypertrophy, intervertebral disc herniation, or the sedimentation sign—are also known to influence CFBS diagnosis and prognosis [26,27,28,29,30,31,32,33]. Incorporating these additional factors may enhance diagnostic accuracy and refine treatment recommendations. Finally, the retrospective, single-center design of this study may introduce selection bias and limit the generalizability of the findings.

Despite these limitations, our findings offer meaningful clinical insights that may support practical applications in diagnosis and treatment planning. The diagnostic cut-off of 33.02 mm^2^ may have practical implications in several clinical contexts. In cases of ambiguous or borderline radicular symptoms, a CFCSA below this threshold could support earlier non-operative interventions. Conversely, confirmation of severe narrowing based on this cut-off may strengthen the indication for surgical decompression, particularly in patients with refractory symptoms. This morphometric parameter may also help avoid overtreatment in patients whose symptoms are unlikely to be explained by cervical foraminal narrowing. Overall, this cut-off value may serve as a useful tool for multidisciplinary decision-making and patient counseling.

In conclusion, this study marks an important step toward establishing CFCSA as a key diagnostic parameter for CFBS. By identifying an optimal CFCSA threshold, we offer clinicians a valuable tool to more accurately diagnose CFBS and distinguish it from other conditions presenting with similar neuropathic symptoms. Although further research is required to confirm and expand upon our findings, the present results underscore the potential of CFCSA measurements to improve the clinical management of CFBS, ultimately facilitating more targeted interventions and better patient outcomes.

To build upon the current findings, future investigations should address several limitations noted in this study, including the relatively small sample size, the exclusive focus on the C5/6 level, and the lack of CT-based differentiation of underlying anatomical causes. Expanding the analysis to include other cervical levels, broader patient demographics, and additional structural parameters may enhance the generalizability and diagnostic utility of CFCSA. Furthermore, although we did not assess occupational history or physical activity profiles, the potential role of cumulative cervical loading due to repetitive occupational or daily activities cannot be overlooked. Professions involving sustained neck postures or mechanical strain may contribute to degenerative foraminal changes. Future prospective studies incorporating occupational data and activity metrics could clarify whether CFBS should be partially understood as a work-related degenerative condition, with important implications for both diagnosis and prevention in occupational health. Lastly, advanced imaging techniques such as 3D volumetric MRI or CT-based reconstructions may offer more detailed assessments of foraminal morphology. The integration of artificial intelligence (AI)-based segmentation tools may further enhance measurement reproducibility and enable large-scale diagnostic applications.

## 5. Conclusions

Lower CFCSA values demonstrated a strong correlation with the likelihood of CFBS, suggesting that as the cross-sectional area diminishes, the risk of foraminal narrowing and ensuing nerve root compromise increases. Through our analysis, we identified 33.02 mm^2^ as the most appropriate cut-off threshold, a value that yielded an impressive sensitivity of 86.4% and a specificity of 86.7%. This particular threshold offers a clinically important value for distinguishing individuals who may be affected by CFBS from those who are not. By integrating this cut-off value into routine diagnostic evaluations, clinicians could more accurately identify patients at risk of CFBS and tailor their treatment strategies accordingly, ultimately improving patient care and outcomes. Given its retrospective, single-center design, the findings of this study require confirmation through prospective, multicenter research before broader clinical adoption.

## Figures and Tables

**Figure 1 tomography-11-00067-f001:**
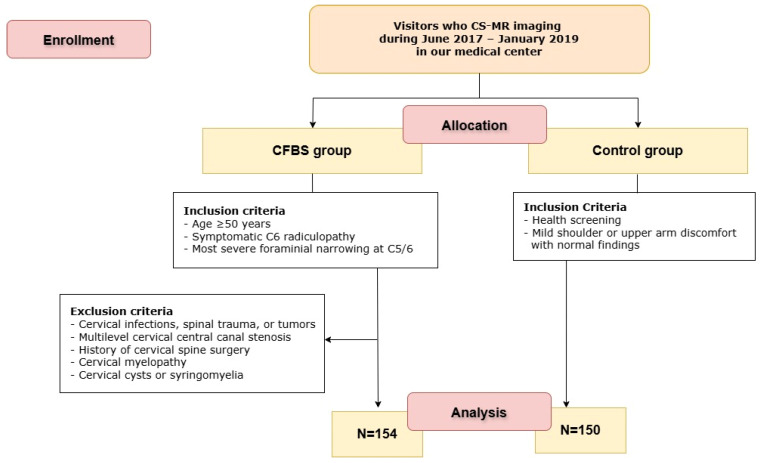
Flow diagram of the study. CS-MR CFBS = cervical spine magnetic resonance imaging; CFBS = cervical foraminal bony stenosis.

**Figure 2 tomography-11-00067-f002:**
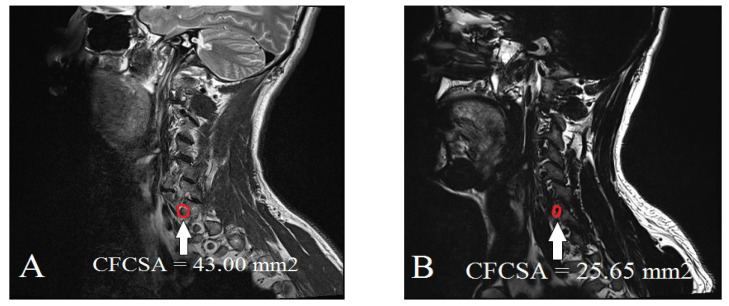
Measurement of the cervical foraminal cross-sectional area on CS-MR at C5/6 stenotic level: (**A**) control group; (**B**) cervical foraminal stenosis group.

**Figure 3 tomography-11-00067-f003:**
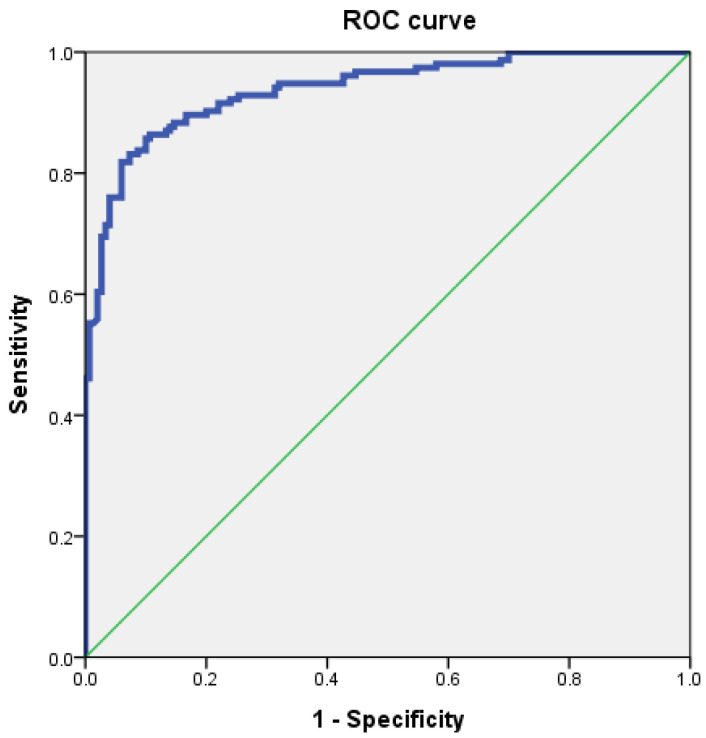
Receiver operating characteristic (ROC) curve of cervical foraminal cross-sectional area (CFCSA) for prediction of cervical foraminal stenosis. The best cut-off point of CFCSA was 33.02 mm^2^, with sensitivity 86.4%, specificity 86.7%, and AUC 0.94.

**Table 1 tomography-11-00067-t001:** Comparison of age, sex, and cervical CFCSA between control and CFBS groups.

Variable	Control Group (*n* = 150)	CFBS Group (*n* = 154)	Statistical Significance
Gender (male/female)	52/98	94/60	NS
Age (yrs)	57.19 ± 6.29	58.53 ± 7.29	NS
CFCSA (mm^2^)	43.00 ± 8.38	25.65 ± 7.19	*p* < 0.001

Data represent the mean ± standard deviation (SD) or the numbers of patients. CFBS = cervical foraminal bony stenosis; CFCSA = cervical foraminal cross-sectional area; NS = not statistically significant (*p* > 0.05).

**Table 2 tomography-11-00067-t002:** Age distribution of patients with mean CFCSA of control group and CFBS group.

Age Distribution (Years)	Total (*n*)		CFCSA (mm^2^)	
	Control	CFBS	Control	CFBS
50–59	104	104	42.71 ± 8.42	24.86 ± 7.10
60–69	40	34	44.41 ± 7.78	26.59 ± 7.51
70–79	6	16	38.67 ± 10.96	28.75 ± 6.32

CFCSA = cervical foraminal joint cross-sectional area; CFBS = cervical foraminal bony stenosis.

## Data Availability

The data generated in this study are available upon request from the corresponding author.

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
