# Peer review of "Determination of the Most Suitable Cut-Off Point of the Cervical Foraminal Cross-Sectional Area at the C5/6 Level to Predict Cervical Foraminal Bony Stenosis"

_tomography, 2025, doi:10.3390/tomography11060067_

Round 1
Reviewer 1 Report
Comments and Suggestions for Authors
The abstract does not clearly state the main objective or purpose of the study. It mentions evaluating the relationship between CFBS and foraminal narrowing, but the main goal of identifying an optimal CFCSA cutoff for diagnosing CFBS should be stated upfront. The study design and methodology are not sufficiently summarized. Details like the retrospective review, inclusion/exclusion criteria, imaging protocols, and statistical analyses used should be briefly described.
The key results and conclusions are mentioned but could be stated more clearly and concisely, focusing on the main findings regarding the CFCSA cutoff value, sensitivity/specificity, etc. Some additional limitations of the study should be acknowledged in the abstract, such as the single-center retrospective design and focus on only the C5/6 level. The introduction provides good background on CFBS but does not clearly build a case for the need to establish a CFCSA cutoff for diagnosis. This rationale should be further developed.
More details are needed on prior grading systems for CFBS and their limitations to highlight the gap this study aims to address.
The specific objective and hypothesis of investigating CFCSA as a diagnostic parameter should be clearly stated at the end of the introduction.
The methods generally describe the study design, patient population, imaging protocols, and statistical analysis appropriately. However, some parts require further clarification or detail like the inclusion/exclusion criteria that should be presented more systematically in a structured list format.The MRI parameters (slice thickness, TR, TE etc) should be specified for both T2 sagittal and axial images. More information is needed on how CFCSA measurements were obtained - the specific anatomic landmarks used to delineate the foraminal area.
The statistical tests used to compare CFCSA between groups and age-related differences should be specified. The results section sufficiently covers the key findings but could be organized more logically like begin with the demographic and clinical characteristics of the patients and controls before presenting CFCSA data. Present CFCSA differences between groups before the age-related data and ROC analysis. The discussion Interprets the results well and acknowledges some limitations. Comparing the study's CFCSA cutoff to thresholds suggested by previous similar studies. Describing specific future research directions beyond general statements.
The conclusions accurately summarize the key findings regarding the CFCSA cutoff value. However, some additional limitations should be acknowledged, such as the retrospective single-center design.
Author Response
- Abstract: The abstract does not clearly state the main objective or purpose of the study. It mentions evaluating the relationship between CFBS and foraminal narrowing, but the main goal of identifying an optimal CFCSA cutoff for diagnosing CFBS should be stated upfront. The study design and methodology are not sufficiently summarized. Details like the retrospective review, inclusion/exclusion criteria, imaging protocols, and statistical analyses used should be briefly described.
The key results and conclusions are mentioned but could be stated more clearly and concisely, focusing on the main findings regarding the CFCSA cutoff value, sensitivity/specificity, etc. Some additional limitations of the study should be acknowledged in the abstract, such as the single-center retrospective design and focus on only the C5/6 level.
Response:
Thank you for the helpful suggestion. We revised the abstract to explicitly state the primary objective: identifying an optimal CFCSA cut-off value for diagnosing CFBS. Additionally, we added a concise summary of the retrospective design, patient selection, imaging analysis, and statistical methods.
Revised Abstract
Background: Cervical foraminal bony stenosis (CFBS) is a common degenerative spinal condition that causes radicular pain and functional impairment in the upper extremities. Accurate and objective diagnosis of CFBS remains challenging due to the absence of standardized morphometric criteria. This study aimed to determine an optimal cut-off value for the cervical foraminal cross-sectional area (CFCSA) at the C5/6 level as a diagnostic indicator of CFBS. Methods: We conducted a retrospective case-control study including 154 patients aged 50 years or older with clinically and radiologically confirmed CFBS and 150 age-matched asymptomatic controls. Cervical spine magnetic resonance imaging (MRI) was performed in all subjects, and CFCSA measurements were obtained from sagittal T2-weighted images using a standardized protocol. Group differences were analyzed using t-tests, and diagnostic performance was assessed using receiver operating characteristic (ROC) curve analysis. Results: The mean CFCSA was significantly lower in the CFBS group (25.65 ± 7.19 mm²) compared to the control group (43.00 ± 8.38 mm²; p < 0.001). ROC analysis identified a CFCSA threshold of 33.02 mm² as the optimal cut-off point for predicting CFBS, yielding a sensitivity of 86.4%, a specificity of 86.7%, and an area under the curve (AUC) of 0.94 (95% CI: 0.91–0.96). Conclusions: These findings suggest that CFCSA is a robust and reproducible morphological parameter for evaluating foraminal stenosis. The proposed cut-off may enhance diagnostic accuracy and aid in clinical decision-making for patients presenting with C6 radiculopathy. However, further validation through multicenter, prospective studies across multiple cervical levels is warranted.
- Introduction (1) : The introduction provides good background on CFBS but does not clearly build a case for the need to establish a CFCSA cutoff for diagnosis. This rationale should be further developed.
Response:
We are grateful for your constructive suggestion. We added sentences emphasizing the diagnostic gap and the need for a quantifiable cut-off to standardize CFBS assessment, which is often based on subjective imaging interpretation.
[1. Introduction] (page 3, line 97)
“Despite multiple grading systems, the lack of a quantitative diagnostic criterion remains a challenge in clinical settings. Establishing a validated CFCSA threshold may reduce diagnostic subjectivity and support more standardized evaluation.”
- Introduction (2) : More details are needed on prior grading systems for CFBS and their limitations to highlight the gap this study aims to address.
Response:
We expanded the discussion of existing grading systems and clarified how our study fills this gap.
[1. Introduction] (page 3, line 115)
“Grading systems, such as those proposed by Kim et al. and Park et al., primarily assess relative nerve root width or the extent of perineural fat obliteration, without providing a direct quantification of foraminal space. Furthermore, these methods are often subject to considerable interobserver variability, which may limit their broader clinical applicability [4,5].”
- Introduction (3) : The specific objective and hypothesis of investigating CFCSA as a diagnostic parameter should be clearly stated at the end of the introduction.
Response:
Thank you for this valuable comment. We have revised the final paragraph of the Introduction to clearly articulate both the specific objective and the underlying hypothesis of our study.
[1. Introduction] (page 3, line 121)
“This study aimed to assess the diagnostic utility of CFCSA in identifying CFBS and to establish a clinically applicable cut-off value. We hypothesized that CFCSA would serve as a reliable morphological marker to differentiate CFBS from normal anatomy.”
- 5. Method (1) : The methods generally describe the study design, patient population, imaging protocols, and statistical analysis appropriately. However, some parts require further clarification or detail like the inclusion/exclusion criteria that should be presented more systematically in a structured list format.
Response:
Thank you for your advice. As requested, the "Methods – Patient Selection" section has been revised into a structured list format, without altering the original content, as follows:
[2. Materials and Methods]-[2.1. Patient Selection] (page 4, line 147)
In order to focus our analysis on a well-defined and clinically relevant subgroup, we applied the following inclusion criteria:
(1) Age ≥ 50 years
(2) Symptomatic C6 radiculopathy based on clinical evaluation
(3) Most severe foraminal narrowing observed at the C5/6 level on MRI
(4) Underwent cervical spine MRI (CS-MR) within 6 months of initial CFBS diagnosis
To ensure the reliability and clinical relevance of our findings, we also applied the following exclusion criteria, aiming to eliminate confounding conditions or anatomical variations that could impact imaging interpretation:
(1) Presence of cervical infections, trauma, or tumors
(2) Moderate to severe central canal stenosis at cervical levels other than C5/6
(3) History of cervical spine surgery
(4) Evidence of cervical myelopathy
(5) Presence of cervical cysts or syringomyelia
We believe that this revision enhances the overall presentation and ensures consistency with common reporting practices in diagnostic imaging research. Thank you again for the helpful feedback.
- Method (2): The MRI parameters (slice thickness, TR, TE etc) should be specified for both T2 sagittal and axial images.
Response:
Thank you for this important comment. In the originally submitted manuscript, we provided detailed imaging parameters for sagittal T2-weighted images but did not fully specify those for axial T2-weighted images. We have now revised the Imaging Protocols section to include acquisition details for both sagittal and axial T2-weighted sequences.
[2. Materials and Methods]-[ 2.2. Imaging Protocols] (page 5, line 185)
“Sagittal T2-weighted images were obtained with the following parameters: slice thickness = 2.0 mm, interslice gap = 0.4 mm, repetition time (TR) = 3740 ms, echo time (TE) = 87 ms, field of view (FOV) = 240 × 240 mm, matrix size = 448 × 314, and echo train length (ETL) = 15.
Axial T2-weighted images were acquired with the following parameters: slice thickness = 3.0 mm, interslice gap = 0.3 mm, TR = 3000–4000 ms, TE = 90–120 ms, FOV = 180 × 180 mm, and matrix size = 320 × 256.”
We appreciate the reviewer’s attention to detail, which has allowed us to improve the methodological transparency and reproducibility of the imaging protocol.
- Method (3): More information is needed on how CFCSA measurements were obtained - the specific anatomic landmarks used to delineate the foraminal area.
Response:
Thank you for this important and constructive comment. In response, we have revised the Image Measurements subsection to provide a more detailed description of the anatomical boundaries and technical approach used for measuring the cervical foraminal cross-sectional area (CFCSA) on MRI.
[2. Materials and Methods]-[ 2.2. Imaging Protocols] (page 5, line 205)
“Specifically, the CFCSA was delineated on sagittal T2-weighted images by manually tracing the bony margins of the neural foramen at the C5/6 level, using the following anatomical landmarks:
Superior boundary: Inferior margin of the C5 pedicle
Inferior boundary: Superior margin of the C6 pedicle
Anterior boundary: Posterior vertebral body and uncovertebral joint
Posterior boundary: Facet joint complex and lamina
To enhance accuracy and ensure appropriate level localization, all sagittal measurements were cross-referenced with axial T2-weighted images. Only slices in which the full vertical extent of the foramen was clearly visible without motion artifacts were used for analysis. Two experienced neuroradiologists performed the measurements independently using the region-of-interest (ROI) tracing tool within the INFINITT PACS system.”
We believe this addition improves the reproducibility and anatomical clarity of the measurement technique, and we are grateful to the reviewer for prompting this important clarification.
- Method (4) : The statistical tests used to compare CFCSA between groups and age-related differences should be specified.
Response:
We appreciate your insightful comment. We have revised the manuscript to explicitly specify the statistical methods used for group comparisons and age-related analysis.
[2. Materials and Methods]-[ 2.4. Statistical Analysis] (page 6, line 224)
“To compare CFCSA values between the CFBS group and the asymptomatic control group, an unpaired t-test was performed. The threshold for statistical significance was set at p < 0.05. To assess age-related differences in CFCSA within the overall study population, subjects were stratified into three age groups: 50–59, 60–69, and ≥70 years. One-way analysis of variance (ANOVA) was used to compare CFCSA measur ements across these age groups.”
- Results : The results section sufficiently covers the key findings but could be organized more logically like begin with the demographic and clinical characteristics of the patients and controls before presenting CFCSA data. Present CFCSA differences between groups before the age-related data and ROC analysis.
Response:
Thank you for your helpful suggestion. We have revised the Results section to follow a more logical structure as suggested. Demographic characteristics (age and sex) are now presented first, followed by CFCSA group comparison, age-stratified analysis, and ROC analysis. The title of Table 1 was also updated to accurately reflect its contents, and CFCSA was visually separated to enhance clarity.
“The baseline characteristics of the control and CFBS groups are presented in Table 1. There were no statistically significant differences in demographic characteristics between the groups. The mean CFCSA was significantly smaller in the CFBS group (25.65 ± 7.19 mm²) compared to the control group (43.00 ± 8.38 mm², p < 0.001).”
Table 1. Comparison of age, sex, and cervical CFCSA between control and CFBS groups
Variable |
Control Group (n = 150) |
CFBS Group (n = 154) |
Statistical significance |
Gender (male/female) |
52/98 |
94/60 |
NS |
Age (yrs) |
57.19 ± 6.29 |
58.53 ± 7.29 |
NS |
CFCSA (mm2) |
43.00 ± 8.38 |
25.65 ± 7.19 |
p < 0.001 |
Data represent the mean ± standard deviation (SD) or the numbers of patients. CFBS = cervical foraminal bony stenosis ; CFCSA = cervical foraminal cross-sectional area; NS = not statistically significant (p > 0.05).
|
- Discussion (1): The discussion Interprets the results well and acknowledges some limitations. Comparing the study's CFCSA cutoff to thresholds suggested by previous similar studies.
Response:
Thank you for this suggestion. In response, we added a discussion paragraph comparing our results with those of An et al., who proposed a cut-off value based on cervical facet joint area (CFJA), a structurally different parameter. While their study shared a similar ROC-based approach, our results demonstrated superior diagnostic performance using CFCSA, which more directly reflects foraminal narrowing. We believe this comparison helps clarify the novelty and clinical utility of CFCSA as a diagnostic indicator for CFBS.
[4. Discussion] (page 10, line 318)
“The primary morphological parameter in our study, the cervical foraminal cross-sectional area (CFCSA), demonstrated a statistically significant difference be-tween groups, with a proposed cut-off value of 33.02 mm². This parameter showed strong diagnostic performance in identifying CFBS, with a sensitivity of 86.4%, specificity of 86.7%, and an AUC of 0.94 (95% CI: 0.91–0.96). In comparison, An et al. proposed a cut-off value of 113.14 mm² for the cervical facet joint area (CFJA) to predict foraminal stenosis in patients aged 50 and older at the C5/6 level, reporting a sensitivity of 70.6%, specificity of 68.6%, and an AUC of 0.72 (95% CI: 0.66–0.77) [15]. While both studies adopted a cross-sectional area–based diagnostic approach using ROC curve analysis, their measurement focused on facet hypertrophy rather than direct evaluation of foraminal narrowing. Moreover, the diagnostic performance of CFCSA in our study was notably superior, reinforcing its potential utility as a more precise and reproducible indicator of CFBS.
- Discussion (2): Describing specific future research directions beyond general statements.
Response:
Thank you for your insightful comment. In response, we have revised the final paragraph of the Discussion section to provide more specific and actionable directions for future research.
[4. Discussion] (page 10, line 387)
“To build upon the current findings, future investigations should address several limitations noted in this study, including the relatively small sample size, the exclusive focus on the C5/6 level, and the lack of CT-based differentiation of underlying anatomical causes. Expanding the analysis to include other cervical levels, broader patient demographics, and additional structural parameters may enhance the generalizability and diagnostic utility of CFCSA.”
- Conclusions : The conclusions accurately summarize the key findings regarding the CFCSA cutoff value. However, some additional limitations should be acknowledged, such as the retrospective single-center design.
Response:
We appreciate the reviewer’s suggestion to elaborate on future research directions in a more specific and actionable manner. In response, we have added a sentence in Conclusions section to include detailed recommendations for future investigations based on the current study’s findings and limitations.
[5.Conclusions] (page 10, line 413)
“Although limited by its retrospective single-center design, this study provides a foundation for future validation and broader clinical application.”
We believe that these specific directions will help guide future research and contribute to the development of more effective diagnostic and therapeutic strategies for CFBS.
We sincerely appreciate all of your insightful comments, which have significantly contributed to improving the clarity and quality of our manuscript.

Reviewer 2 Report
Comments and Suggestions for Authors
The manuscript presents a well-structured and clinically relevant study that seeks to establish a diagnostic cut-off value for the cervical foraminal cross-sectional area (CFCSA) to predict cervical foraminal bony stenosis (CFBS). The research addresses a pertinent gap in spinal diagnostics and offers clear and statistically robust results. The article is written in a clear and scientific manner, and the findings hold meaningful implications for clinical practice. The retrospective case-control design is appropriate for the aim of the study. The inclusion and exclusion criteria are well defined, and the focus on the C5/6 level is justified based on its clinical relevance. The results are clearly presented with well-structured tables and figures. The statistical analysis is thorough and appropriate, with the ROC curve analysis being particularly convincing.
I have the following suggestions:
- Introduction
Consider integrating a brief summary of existing quantitative benchmarks used in similar anatomical diagnostic research. This would better contextualize the novelty of the proposed cut-off value.
- Methods
While the CFCSA measurement process is well explained, more detail on how disagreements between observers were resolved would further strengthen reproducibility. Additionally, incorporating other anatomical variables such as body mass index, gender differences, or vertebral body dimensions could provide a more comprehensive understanding of CFCSA variations.
- Conclusions
The conclusions are consistent with the presented data and supported by a solid discussion of both the strengths and limitations of the study. The discussion could be further enhanced by suggesting specific clinical scenarios where this cut-off value could directly influence therapeutic decision-making (e.g., surgical planning vs conservative treatment).
Author Response
- Introduction : Consider integrating a brief summary of existing quantitative benchmarks used in similar anatomical diagnostic research. This would better contextualize the novelty of the proposed cut-off value.
Response:
Thank you for the insightful suggestion. We have revised the Introduction to emphasize the absence of established CFCSA thresholds in prior research, thereby highlighting the novelty and clinical relevance of our proposed cut-off value.
[1. Introduction] (page 3, line 97)
“Despite multiple grading systems, the lack of a quantitative diagnostic criterion remains a challenge in clinical settings. Establishing a validated CFCSA threshold may reduce diagnostic subjectivity and support more standardized evaluation.”
[1. Introduction] (page 3, line 115)
“Grading systems, such as those proposed by Kim et al. and Park et al., primarily assess relative nerve root width or the extent of perineural fat obliteration, without providing a direct quantification of foraminal space.”
- Method (1) : While the CFCSA measurement process is well explained, more detail on how disagreements between observers were resolved would further strengthen reproducibility
Response:
We have added a statement to address how discrepancies between observers were resolved.
[2. Materials and Methods]-[2.3. Image Measurements] (page 6, line 218)
“Discrepancies exceeding 5% in CFCSA measurements were resolved by joint review and consensus between the two neuroradiologists.”
- Mehotd (2): Additionally, incorporating other anatomical variables such as body mass index, gender differences, or vertebral body dimensions could provide a more comprehensive understanding of CFCSA variations.
Response:
Thank you for your thoughtful suggestion. We agree that anatomical variables such as body mass index (BMI), gender, and vertebral body dimensions could provide valuable insights into CFCSA variations. Although these variables were not available in our dataset and were not included in the present analysis, we had already acknowledged this as a study limitation in the Discussion section. The relevant sentence is as follows:
[4. Discussion] (page 10, line 360)
“Fifth, other demographic factors, such as gender, body size, or body mass index, were not analyzed in this study. These factors may potentially influence CFCSA and warrant further investigation.”
We hope this sufficiently addresses your comment.
- Discussion : The discussion could be further enhanced by suggesting specific clinical scenarios where this cut-off value could directly influence therapeutic decision-making (e.g., surgical planning vs conservative treatment).
Response:
We appreciate this valuable suggestion. To address it, we have briefly expanded the Discussion to highlight how the proposed CFCSA cut-off value may assist in real-world clinical decisions. Specifically, we mention its potential role in supporting earlier conservative treatment in borderline cases, reinforcing surgical indications in refractory symptoms, and avoiding unnecessary interventions. These additions aim to illustrate the clinical applicability of our findings without requiring major structural changes or additional references.
Accordingly, the following paragraph has been added to the revised Discussion section:
[4. Discussion] (page 10, line 368)
“Despite these limitations, our findings offer meaningful clinical insights that may support practical applications in diagnosis and treatment planning. The diagnostic cut-off of 33.02 mm² may have practical implications in several clinical contexts. In cases of ambiguous or borderline radicular symptoms, a CFCSA below this threshold could support earlier non-operative interventions. Conversely, confirmation of severe narrowing based on this cut-off may strengthen the indication for surgical decompression, particularly in patients with refractory symptoms. This morphometric parameter may also help avoid overtreatment in patients whose symptoms are unlikely to be explained by cervical foraminal narrowing. Overall, this cut-off value may serve as a useful tool for multidisciplinary decision-making and patient counseling.”
We are grateful to the reviewer for highlighting the importance of contextualizing our findings, and we believe this addition enhances the translational value of our study.

Reviewer 3 Report
Comments and Suggestions for Authors
1. The manuscript addresses a clinically relevant gap by establishing a quantitative diagnostic threshold for cervical foraminal bony stenosis at C5/6. This provides standardized criteria for CFBS diagnosis that can potentially reduce subjectivity in radiological assessments.
2. The literature review is comprehensive and current. However, the authors could benefit from including comparative studies using 3D imaging or AI based morphological assessments, which are emerging in spine diagnostics.
3. The methodology is clear and reproducible - the inclusion and exclusion criteria are detailed, measurement protocols are well described, and imaging parameters are provided.
4. The retrospective nature and demographic limitations (age > 50, C6 radiculopathy only) should be acknowledged more explicitly in methods rather than only in discussion.
5. Results are clearly and thoroughly presented, but stratification by age shows no significant difference in CFCSA, but the small sample size in older age groups (e.g., only 6 in age group 70–79) limits subgroup analysis strength. Discuss the limitation imposed by this small sample.
The paper is strong and relevant but would benefit from minor enhancements in discussing generalizability, and practical implementation more details.
Author Response
- The manuscript addresses a clinically relevant gap by establishing a quantitative diagnostic threshold for cervical foraminal bony stenosis at C5/6. This provides standardized criteria for CFBS diagnosis that can potentially reduce subjectivity in radiological assessments.
Response:
We sincerely thank the reviewer for recognizing the clinical relevance and potential impact of our study.
- The literature review is comprehensive and current. However, the authors could benefit from including comparative studies using 3D imaging or AI based morphological assessments, which are emerging in spine diagnostics.
Response:
We sincerely thank the reviewer for this forward-looking and highly relevant suggestion. The integration of artificial intelligence (AI) and advanced 3D imaging techniques into spinal diagnostics represents a rapidly evolving area with great potential to enhance the precision, reproducibility, and scalability of morphometric assessments such as CFCSA.
In response, we have added a paragraph to the Discussion section addressing how future studies could incorporate AI-based segmentation tools or 3D volumetric reconstruction from MRI or CT data to further refine the evaluation of cervical foraminal stenosis. These methods may help reduce interobserver variability and allow for automated screening in large patient cohorts.
[4. Discussion] (page 10, line 399)
“Lastly, advanced imaging techniques such as 3D volumetric MRI or CT-based reconstructions may offer more detailed assessments of foraminal morphology. The integration of artificial intelligence (AI)-based segmentation tools may further enhance measurement reproducibility and enable large-scale diagnostic applications.”
We greatly appreciate the reviewer’s suggestion, which we believe has meaningfully strengthened the translational relevance and future orientation of our study.
- The methodology is clear and reproducible - the inclusion and exclusion criteria are detailed, measurement protocols are well described, and imaging parameters are provided.
Response:
Thank you for your kind comment regarding the clarity of our methods.
- Method: The retrospective nature and demographic limitations (age > 50, C6 radiculopathy only) should be acknowledged more explicitly in methods rather than only in discussion.
Response:
Thank you for this helpful comment. The retrospective design, as well as the inclusion restrictions based on age (≥50 years) and C6 radiculopathy, were already described in the original version. However, to improve clarity and alignment with the reviewer’s suggestion, we have reorganized and explicitly highlighted these criteria in the Methods section.
[2. Materials and Methods]-[2.1. Patient Selection] (page 4, line 147)
“ In order to focus our analysis on a well-defined and clinically relevant subgroup, we applied the following inclusion criteria:
(1) Age ≥ 50 years
(2) Symptomatic C6 radiculopathy based on clinical evaluation
(3) Most severe foraminal narrowing observed at the C5/6 level on MRI
(4) Underwent cervical spine MRI (CS-MR) within 6 months of initial CFBS diagnosis
To ensure the reliability and clinical relevance of our findings, we also applied the following exclusion criteria, aiming to eliminate confounding conditions or anatomical variations that could impact imaging interpretation:
(1) Presence of cervical infections, trauma, or tumors
(2) Moderate to severe central canal stenosis at cervical levels other than C5/6
(3) History of cervical spine surgery
(4) Evidence of cervical myelopathy
(5) Presence of cervical cysts or syringomyelia “
- Results : Results are clearly and thoroughly presented, but stratification by age shows no significant difference in CFCSA, but the small sample size in older age groups (e.g., only 6 in age group 70–79) limits subgroup analysis strength. Discuss the limitation imposed by this small sample.
Response:
Thank you for this insightful comment. We acknowledge this limitation and have added a sentence in the Discussion to address it.
[4. Discussion] (page 9, line 346)
“In particular, the small number of participants in the older age subgroups (e.g., 70–79 years) limits the statistical power of age-stratified analyses and may have contributed to the lack of significant findings in age-related comparisons.”

Reviewer 4 Report
Comments and Suggestions for Authors
Interesting study that has as its aim the study of cervical foraminal bone stenosis. Considering the number of patients who come to the clinic for pathologies of different nature and among these a conspicuous part presents a symptomatology that has nothing to do with the reason why they came to our observation, we can understand the importance that the pathology studied by our research colleagues is present in the population and represents an important fact. Excellent development of the study project and the choice of subjects to study, one is struck in reading the paper of how many have chosen to be examined with an exam, the MRI that requires long times, noises to be tolerated, a generally cold environment. The results are in line with what was hypothesized in the study project. An important fact that we ask colleagues and there is a hypothetical relationship with the daily activities carried out by the people who presented such symptoms and if the cervical foraminal bone stenosis could therefore be interpreted as an occupational pathology. This could have important implications for occupational medicine, all this considering also the age range taken into consideration by the speakers of the article. Good iconography, good English, good bibliography
Author Response
Interesting study that has as its aim the study of cervical foraminal bone stenosis. Considering the number of patients who come to the clinic for pathologies of different nature and among these a conspicuous part presents a symptomatology that has nothing to do with the reason why they came to our observation, we can understand the importance that the pathology studied by our research colleagues is present in the population and represents an important fact. Excellent development of the study project and the choice of subjects to study, one is struck in reading the paper of how many have chosen to be examined with an exam, the MRI that requires long times, noises to be tolerated, a generally cold environment. The results are in line with what was hypothesized in the study project. An important fact that we ask colleagues and there is a hypothetical relationship with the daily activities carried out by the people who presented such symptoms and if the cervical foraminal bone stenosis could therefore be interpreted as an occupational pathology. This could have important implications for occupational medicine, all this considering also the age range taken into consideration by the speakers of the article. Good iconography, good English, good bibliography.
Response:
We greatly appreciate the reviewer’s thoughtful and insightful comment regarding the potential link between daily activities or occupational exposures and the development of cervical foraminal bony stenosis (CFBS). This is indeed an important point that could have significant implications for occupational medicine and preventive spine care, particularly in aging populations.
While our current study was not designed to directly evaluate occupational factors, we fully agree that the mechanical loading and repetitive motion associated with certain occupations may play a critical role in the development and progression of CFBS. Previous biomechanical and epidemiological studies have shown that cumulative cervical spine stress, particularly in professions involving repetitive neck extension, flexion, axial loading, or prolonged static postures (e.g., assembly line workers, dental professionals, manual laborers), may accelerate degenerative changes in the cervical intervertebral joints and foramina. These changes include facet joint hypertrophy, osteophyte formation, and disc space narrowing—all of which can contribute to foraminal narrowing and ultimately nerve root impingement.
In our study, we included patients aged 50 and above, a demographic in which cumulative mechanical stress is likely to have played a significant role over decades of occupational and daily activity. Although we did not collect detailed occupational histories or stratify subjects based on job type or activity level, we agree that such an analysis would provide valuable insight. In particular, investigating whether individuals in physically demanding or posture-intensive occupations demonstrate smaller cervical foraminal cross-sectional areas (CFCSA) or earlier onset of CFBS would help to better understand the etiologic pathways of the disease.
To acknowledge the reviewer’s point, we have added the following text to the Discussion section of the revised manuscript:
[4. Discussion] (page 10, line 392)
“Furthermore, although we did not assess occupational history or physical activity profiles, the potential role of cumulative cervical loading due to repetitive occupational or daily activities cannot be overlooked. Professions involving sustained neck postures or mechanical strain may contribute to degenerative foraminal changes. Future prospective studies incorporating occupational data and activity metrics could clarify whether CFBS should be partially understood as a work-related degenerative condition, with important implications for both diagnosis and prevention in occupational health.”
We again thank the reviewer for raising this important consideration, and we hope that our response and corresponding revision reflect the relevance of their insight.

Round 2
Reviewer 1 Report
Comments and Suggestions for Authors
The abstract conclusions state that further validation is needed, but do not explicitly mention the limitations of the single center retrospective design of this study. This limitation should be acknowledged.The introduction section provides extensive background on CFBS but does not clearly identify the knowledge gap this study aims to address regarding the lack of a standardized diagnostic parameter. This gap and objective should be more clearly stated.The MRI protocols are described but important parameters like field strength are missing. These details should be provided.
The statistical analysis section does not mention the tests used to assess normality of data distribution prior to t-tests. This information is needed.
The results report various statistical values but do not mention whether tests for normality were conducted. This is an important methodological detail.
When presenting the AUC, the standard error and 95% CI should also be reported for completeness.In the discussion section the implications of the small sample size, particularly in older subgroups, should be addressed as it limits the conclusions.
The discussion emphasizes the need for multicenter prospective validation but does not highlight the limitations of the single center retrospective design of this study. This limitation needs to be directly acknowledged.The conclusion summarizes the main findings but does not reiterate the need for further multicenter validation due to the limitations of the current study design. This message should be reinforced.
Author Response
We sincerely thank the reviewer for the insightful and constructive comments. We have carefully addressed each point and revised the manuscript accordingly. The corresponding changes have been clearly marked with underlines in the revised manuscript. Below are our point-by-point responses.
- The abstract conclusions state that further validation is needed, but do not explicitly mention the limitations of the single center retrospective design of this study. This limitation should be acknowledged.
Thank you for this important point. We have slightly revised the Abstract to acknowledge the retrospective, single-center design as a limitation, as requested.
“However, given its retrospective, single-center design, further validation through multicenter, prospective studies across multiple cervical levels is warranted.” (page1, line 34)
- The introduction section provides extensive background on CFBS but does not clearly identify the knowledge gap this study aims to address regarding the lack of a standardized diagnostic parameter. This gap and objective should be more clearly stated.
Thank you for this helpful suggestion. In response, we have clarified the knowledge gap regarding the absence of a standardized diagnostic parameter for CFBS and explicitly stated the objective of our study. To address this, the following sentences have been added to the Introduction:
“Despite multiple grading systems, the absence of a validated, quantitative imaging criterion continues to hinder consistent diagnosis. Defining such a parameter could reduce diagnostic subjectivity and facilitate more standardized clinical assessment—particularly through morphometric indicators such as cross-sectional foraminal area.” (page3, line 100)
“This highlights the need to establish a quantitative threshold for CFCSA that can serve as a standardized diagnostic parameter in clinical practice.” (page3, line 124)
- The MRI protocols are described but important parameters like field strength are missing. These details should be provided.
Thank you for this helpful comment. In response to a previous reviewer’s suggestion, we have already added detailed MRI acquisition parameters to the Methods section, including field strength (3T), scanner models, and sequence-specific technical details (e.g., TR, TE, FOV, matrix size, slice thickness). We believe these additions sufficiently address the reviewer’s concern.
[2. Materials and Methods]-[2.2. Imaging Protocols] (page 5, line 187)
“All CS-MR imaging was performed using high-field 3T MRI scanners (Avanto, Siemens Healthcare, Germany; and Philips Achieva, Philips Healthcare, Best, The Netherlands). Sagittal T2-weighted images were obtained with the following parameters: slice thickness = 2.0 mm, interslice gap = 0.4 mm, repetition time (TR) = 3740 ms, echo time (TE) = 87 ms, field of view (FOV) = 240 × 240 mm, matrix size = 448 × 314, and echo train length (ETL) = 15. Axial T2-weighted images were acquired with the following parameters: slice thickness = 3.0 mm, interslice gap = 0.3 mm, TR = 3000–4000 ms, TE = 90–120 ms, FOV = 180 × 180 mm, and matrix size = 320 × 256.”
- The statistical analysis section does not mention the tests used to assess normality of data distribution prior to t-tests. This information is needed. The results report various statistical values but do not mention whether tests for normality were conducted. This is an important methodological detail.
Thank you for the comment. We have revised the Statistical Analysis section to include the normality assessment method used prior to applying t-tests.
“Normality of distribution was assessed using the Shapiro–Wilk test prior to conducting parametric analyses.” (page 6, line 233)
We have also added a statement to the Results section as suggested.
“All continuous variables satisfied the assumption of normality.” (page 6, line 245)
- When presenting the AUC, the standard error and 95% CI should also be reported for completeness.
Thank you for this valuable suggestion. We agree that presenting the 95% confidence interval (CI) and standard error (SE) for the area under the ROC curve (AUC) enhances the statistical rigor and completeness of the results. The AUC for CFCSA in predicting CFBS has already been reported with a 95% CI (0.91–0.96); in response to this comment, we have now added the standard error of the AUC, which is 0.0128, to the Results section and Figure 3 legend:
“The optimal cut-off value for CFCSA was 33.02 mm², with a sensitivity of 86.4%, specificity of 86.7%, and an AUC of 0.94 (standard error = 0.0128; 95% CI, 0.91–0.96).” (page 7, line 266)
- In the discussion section the implications of the small sample size, particularly in older subgroups, should be addressed as it limits the conclusions.
Thank you for this valuable comment. As suggested, we have already addressed this point in the revised Discussion section in response to a previous reviewer’s comment. Specifically, we acknowledged the relatively small sample size and its limitations in age-stratified analysis, particularly within the older subgroups. The relevant sentences are as follows:
“First, the relatively small sample size necessitates further research involving larger cohorts to validate our findings and ensure their generalizability. In particular, the small number of participants in the older age subgroups (e.g., 70–79 years) limits the statistical power of age-stratified analyses and may have contributed to the lack of significant findings in age-related comparisons.” (page 10, line 352)
We hope this satisfactorily addresses the reviewer’s concern.
- The discussion emphasizes the need for multicenter prospective validation but does not highlight the limitations of the single center retrospective design of this study. This limitation needs to be directly acknowledged.
Thank you for this important comment. In response, we have added an explicit statement at the end of the limitations paragraph to directly acknowledge the retrospective, single-center design of the study and its potential impact on generalizability, as suggested.
“Finally, the retrospective, single-center design of this study may introduce selection bias and limit the generalizability of the findings.” (page 10, line 373)
- The conclusion summarizes the main findings but does not reiterate the need for further multicenter validation due to the limitations of the current study design. This message should be reinforced.
Thank you for this insightful comment. In response, we revised the final sentence of the Conclusion to replace the previous general wording with a more explicit and definitive statement.
“Given its retrospective, single-center design, the findings of this study require confirmation through prospective, multicenter research before broader clinical adoption.” (page 11, line 422)
Thank you again for your valuable feedback and guidance throughout the revision process.
